# The Clinical Characteristics and Prognostic Factors of Primary Extramammary Paget’s Disease Treated with Surgery in Anogenital Regions: A Large Population Study from the SEER Database and Our Centre

**DOI:** 10.3390/jcm12020582

**Published:** 2023-01-11

**Authors:** Zeyang Chen, Zining Liu, Shaorong Pan, Jin Liu, Shuai Zuo, Pengyuan Wang

**Affiliations:** 1Department of General Surgery, Peking University First Hospital, Peking University, 8 Xi ShiKu Street, Xicheng District, Beijing 100034, China; 2Department of Gynecology Oncology, National Cancer Center/National Clinical Research Center for Cancer/Cancer Hospital, Chinese Academy of Medical Sciences and Peking Union Medical College, Beijing 100021, China

**Keywords:** extramammary Paget’s disease (EMPD), SEER database, anogenital regions, prognosis, recurrence

## Abstract

Background: Extramammary Paget’s disease (EMPD) is a rare malignant cutaneous tumour that is commonly located in anogenital regions. The diagnosis of the disease is always delayed, and treatment is usually troublesome. This study aims to summarise the clinicopathological characteristics and the risk factors of prognosis for EMPD in anogenital regions, potentially providing evidence for the diagnosis and treatment of anogenital EMPD. Methods: 688 patients were sourced from the Surveillance, Epidemiology and End Results (SEER) program between 1992 and 2021. In total, 176 participants from our centre from between 2011 and 2021 were included to investigate the characteristics and prognosis for EMPD in anogenital regions. Results: From the SEER program data, patient age of 65 years or older, metastasis of lymph nodes, Spanish-Hispanic-Latino race, diameter exceeding 10cm and lesions located anally were revealed as independent risk factors for shorter cancer-specific survival (CSS). However, the data from our centre highlighted that metastasis of lymph nodes and tumours extending through the epidermis are independent risk factors of shortened progression-free survival (PFS) and CSS of anogenital EMPD. Conclusion: This synthesised study revealed that some characteristics are regarded as risk factors for poor clinical prognosis, which have potential value in formulating more normative and effective strategies for patients with EMPD in anogenital regions.

## 1. Introduction

Extramammary Paget’s disease (EMPD) is a rare malignant cutaneous disease with low incidence, accounting for merely 6.5% of all types of Paget’s disease [1]. Unlike secondary EMPD, which usually arises from the intraepithelial spreading of anorectal or urogenital carcinomas [2], primary EMPDs occur mainly in regions rich in apocrine glands [3]. The vulva is regarded as the most common site of primary EMPD [4], followed by the perianal region [2,5]. Due to this disease’s rarity, most studies investigating primary EMPD involve case series with a limited sample size [6,7,8,9]. The primary EMPD in anogenital regions, deemed the most common site, are frequently misdiagnosed as inflammatory or infectious diseases, as they generally appear as symptoms without specificity, such as rash, pruritus, erosion, exudation and erythema [3,10]. Due to the site of pathological changes, patients cannot observe lesions directly in a convenient way and sometimes hesitate to seek medical advice. Surgical excision remains the first choice for treating EMPD [11]. Because preserving anal and genital function is imperative, operations for EMPD in these sites are intractable [12,13]. Characterising and defining clinical features for EMPD in anogenital regions have potentially positive effects on improving the diagnosis and treatment level, consequently reducing misdiagnosis. Equally, the treatment and prognosis analysis is beneficial for optimising management strategies of EMPD in these atypical sites.

Currently, there is still no research focusing on the clinical characteristics and the prognosis of primary EMPD in anogenital regions. This study combines data from the surveillance, epidemiology and results (SEER) database and our centre to analyse the biological behaviour of primary EMPD in genital and perianal sites by using hundreds of samples to provide scientific evidence.

## 2. Materials and Methods

Permission was obtained from the SEER database to acquire the research data without requiring informed consent from patients. Data collection within our centre about EMPD in anogenital regions was approved by the Institutional Ethics Committee of Peking University First Hospital. Verbal informed consent was obtained from all patients.

### 2.1. Data Collection

The SEER database contains cancer incidence and mortality data from 18 population-based registries which represent approximately 30% of the US population. The data were extracted from the SEER database based on a November 2021 submission using the SEER*Stat software version 8.4.0. The search criteria addressed all patients presenting with histologically confirmed extramammary Paget’s disease except for bones, according to the “{Site and Morphology.Site recode—rare tumours}” and “{Site and Morphology.ICD-O-3 Hist/behav}” diagnosed between 1992 and 2021. A total of 1705 patients were identified. One thousand and seventeen patients were excluded for various reasons, with 688 participants remaining for analysis.

The data of participants with EMPD in anogenital regions were extracted from the SEER database. The information about patients’ age, gender, time of diagnosis, nationality, SEER stage, the status of lymph nodes, the diameter of lesions, sites of EMPD and cancer-specific survival (CSS) were included. We included patients who underwent surgery with primary EMPD (confirmed by pathology) located at the anus, penis, scrotum, vagina and vulva between 1992 and 2021. The screening process is shown in Figure 1, and 688 patients’ data concerning clinical and survival information from the SEER program were analysed in our study.

An additional 176 patients who underwent surgery at our centre for primary EMPD lesions located at genital and perianal lesions from January 2011 to December 2021 were added to the SEER participants. The inclusion criteria were: (I) primary EMPD underwent surgery, and oncological pathology was confirmed by two experienced pathologists who specialise in cutaneous tumours; (II) patients have detailed clinicopathological data and follow-up information about CSS and postoperative recurrence, and (III) EMPD located at anogenital regions (anus, penis, scrotum, vagina or vulva). The demographic data included age, body mass index (BMI), gender, American Society of Anesthesiologists (ASA), comorbidity (hypertension, diabetes mellitus, coronary heart disease, chronic obstructive pulmonary disease, cerebrovascular disease or nephropathy), SEER stage, sites of EMPD, diameter of lesions, surgical margin, lymphadenectomy, invasion level (level 1: in situ in the epidermis; level 2: invasion into the papillary dermis; and level 3: invasion into the reticular dermis or subcutaneous tissue), lymphovascular infiltration (LVI), the use of chemotherapy or radiation, reconstruction ways for surgical wound, surgical types [wide local excision (WLE) or Mohs micrographic surgery (MMS)] and intervals between onset of symptoms and treatment. Patients who received only biopsy results were excluded from our research. Consistent with the SEER program, CSS was defined as the length of time from the date of diagnosis to the date of death from the disease. Disease progression was defined as relapse in surgical sites, metastases in regional lymph nodes and distant metastases. Progression-free survival (PFS) rate was measured from the beginning of diagnosis until the condition’s progression or death.

### 2.2. Statistical Analysis

The associations between clinicopathological characteristics and long-term CSS were assessed using the Kaplan–Meier method (log-rank test). Cox proportional hazards models were used to determine hazard ratios (HR) in univariate analyses. Tumour or treatment characteristics for which univariate analysis reached values of *p* < 0.1 were tested in multivariate models. Covariates included in the multivariate Cox proportional hazards models were selected by a backward stepwise regression (BCR) method based on the smallest Akaike information criterion (AIC) value that indicated the minimal loss of prognostic information [14]. For all analyses, *p* < 0.05 was considered statistically significant. Statistical analyses were performed using R 4.2.0 (https://www.r-project.org/, 30 April 2022).

## 3. Results

### 3.1. Data from the SEER Program

The data in Appendix A represents patients’ clinicopathological data derived from the SEER program. The mean patient age is 69.7 years old, with most patients older than 65 (65.55%). Females (72.97%) were more commonly diagnosed with EMPD in genital or perianal regions. Non-Spanish-Hispanic-Latino (92.3%) patients accounted for most of the target population. The majority of patients were diagnosed with a localised stage (78.78%) and, thus, lacked the definite status of lymph nodes (97.53%) and diameters (56.69%). Vulva (71.51%) and scrotum (21.95%) are the most common locations for EMPD in participants of this study.

Univariate and multivariate analyses, detailed in Table 1, were undertaken to explore the risk factors of short CSS for EMPD patients in anogenital regions. Seven factors with a *p* < 0.1 were identified, namely age > 65 years, year of diagnosis after 2005, SEER stage, metastasis of lymph nodes, Spanish-Hispanic-Latino ethnicity, diameters of lesions > 10 cm and anal EMPD. They were incorporated into the multivariate analysis using the backward stepwise regression (BSR) with a minimum Akaike information criterion (AIC) of 127.44, with no covariate excluded. The multivariate analysis revealed that age > 65 years (HR = 8.51, 95%CI 1.67–43.42, *p* = 0.01), metastasis of lymph nodes (HR = 214.24, 95%CI 8.36–5493.13, *p* = 0.001), Spanish-Hispanic-Latino patients (HR = 13.44, 95%CI 2.47–73.19, *p* = 0.003), diameters of lesions > 10cm (HR = 14.35, 95%CI 2.57–80.15, *p* = 0.002) and anal EMPD (HR = 115.38, 95%CI 6.77–1966.79, *p* = 0.001) were strong predictors of poor prognosis for EMPD of specific regions in the SEER program. Kaplan–Meier curves for the relationships between CSS and age, year of diagnosis, the status of lymph nodes, nationality, lesion diameter and EMPD site are presented in Figure 2. 

### 3.2. Participants from Our Centre

The demographic characteristics of the centre cohort are summarised and presented in Appendix A. The mean age and BMI were 65.81 years old and 24.41 Kg/m^2^, respectively. Males (89.2%) comprise the majority of patients, and most (86.36%) had good fundamental health status, defined as ASA 1–2. In agreement with the SEER program, localised stage EMPD (76.14%) accounted for most patients. Interestingly, unlike in the SEER cohort, the scrotum was the most common site (80.11%). MMS was a frequently used surgical option (75.0%), and only a minimal number of patients underwent lymphadenectomy (4.55%). Safe resection margins (81.82%) occurred in the overwhelming majority of patients. Surprisingly, in most cases (63.07%), intervals between onset of symptoms and treatment exceeded two years.

The univariate and multivariate analyses were done to assess the risk factors for shorter PFS and are shown in Table 2. With the *p* value less than 0.1, eight potential factors (SEER stage, metastasis of lymph nodes, invasion level, LVI, the use of chemotherapy, reconstruction way, surgical type and IST > 2 years) were selected for the BSR model (AIC = 293.60). Four predictors (SEER stage, the use of chemotherapy, reconstruction way and surgical type) were incrementally eliminated, as they did not significantly contribute to the model (AIC = 286.86). Ultimately, lymph node metastases were revealed as a strong predictor of recurrence after surgery (HR = 10.33, 95%CI 2.57–41.48, *p* < 0.001), increasing the risk for recurrence by more than sixfold. Tumours extending through the epidermis were also an independent risk factor for shorter PFS (HR = 2.96, 95%CI 1.14–7.7, *p* = 0.026).

Setting the CSS as the study endpoint, the univariate and multivariate analyses (Table 3) revealed positive margins, metastases of lymph nodes, invasion of tumour extending through the epidermis, LVI and reconstruction by free skin flaps were all risk factors (*p* < 0.05) for poor prognosis. The BSR model eliminated three factors (margin, reconstruction way and LVI), leaving metastasis of lymph nodes (HR = 10.33, 95%CI 2.57–41.48, *p* < 0.001) and invasion of tumour extending through the epidermis (HR = 2.96, 95%CI 1.14–7.7, *p* = 0.026) as independent risk factors for short CSS (AIC dropped from 152.68 to 144.33). Kaplan–Meier curves for the relationships between PFS/CSS and lymph node status and invasion levels are presented in Figure 3.

## 4. Discussion

The anogenital region is the predilection site for bacterial infections or inflammatory diseases. Bacterial infections or inflammatory diseases in this region have an excellent prognosis and present with non-life-threatening symptoms, such as perianal eczema, dermatitis and vulvitis. As the anogenital region is the most common site for EMPD, EMPD in the anogenital region is always misdiagnosed as one of the aforementioned benign cutaneous diseases because of similar unspecific symptoms. Most patients experience intervals between the onset of symptoms and treatment exceeding two years, which is consistent with a previous study [15]. Making a systemic conclusion of clinicopathological and prognostic characteristics for EMPD in anogenital regions unquestionably has enormous positive outcomes for treatment for these EMPD patients, which can be intractable due to preservation of anal and genital function. Results from this study can also improve the understanding of this rare disease for colorectal surgeons, urologists and gynaecologists, reducing the misdiagnosis rate of anogenital EMPD. Similar to EMPD from other sites [3,16,17], in the SEER program of our centre, most patients with anogenital EMPD were older adults over 65 years old. It has been universally recognised that the vulva is the most common site for EMPD [3,4]. The data from the SEER program supported this consensus that most anogenital patients were females with lesions located at the vulva. However, results from our centre revealed that males with lesions located at the scrotum were the main population of anogenital EMPD patients. Our institution is famous for its urology and andrology departments; thus, this inconformity could be due to the referral bias. As such, the centre cohort consisted predominantly of EMPD originating from the male genitalia.

The SEER program outcomes in this study were consistent with the previous research [3], which showed that the elderly over 65 years old have poorer prognoses than young people. This is probably a consequence of poorer fundamental health conditions and the shorter life expectancy of aged people. The anogenital EMPD cases diagnosed after 2005 from the SEER database had significantly better prognoses than patients before 2005 (HR = 0.55, 95%CI 0.3–0.99, *p* = 0.048), which might be due to an improved clinical management strategy for EMPD. The greatest tumour dimension was used as a vital part of the staging system for skin cancer because of its close association with prognosis [3]. Anogenital EMPD with a diameter exceeding 10cm was also regarded as the independent risk factor for shorter CSS from the data in the SEER program participants.

In contrast, similar to older studies, results from our centre suggested that tumour size did not correlate significantly with disease progression or survival [18,19,20]. Most patients had anogenital EMPD in the localised SEER stage, whether in the SEER or centre cohort, with tumours confined to the epidermal layer. Lesions with larger diameters might exhibit diffusion of tumour cells along the epidermis horizontally, which could not increase the risk of distant metastases. Lesions in the perianal regions are supposed to strongly predict an unfavourable prognosis after the multivariate analysis using the SEER data. This finding was consistent with previous studies, as perianal regions with little subcutaneous fat were deemed to have a high risk of metastases even in the early tumour phase [3,17,21,22]. Furthermore, considering the specific anatomical characteristics and protection of organ function, operations for perianal EMPD have increased difficulty in achieving a negative margin than lesions in other sites [20]. According to the SEER data, race might be one of the influencing factors for prognosis, with Spanish-Hispanic-Latino people experiencing less CSS. The SEER program participants were mainly from Western countries, with insufficient participants of Asian origins. Consequently, this study combined data from our centre with a relatively large Asian population among the current studies.

Complete resection of the lesion, which is characterised by ill-defined margins and a high recurrence rate, is regarded as the mainstream therapeutic method for EMPD [23,24]. Matsuo et al. indicated that vulvar Paget’s disease with a positive margin correlates with a higher 5-year cumulative local-recurrence rate (35.8% vs. 15.0%, *p* = 0.01) and lower 5-year overall survival rate (72.6% vs. 88.2%, *p* = 0.032) than a negative margin [25]. In contrast, some studies report that negative margins do not reduce the recurrence rate significantly in patients with anogenital EMPD [26]. The current study found that patients with positive margins had significant shorter CSS (HR = 2.61, 95%CI 1.02–6.65, *p* = 0.045) from the univariate analysis data but without significant PFS when compared with patients with negative margins. WLE, which includes resectioning tumours and normal tissue around the lesion, is a classical surgical method for treating EMPD. Unfortunately, WLE leads to a high local recurrence rate, reaching up to 30% [6], and vast tissue loss, resulting in deformity and impaired function in anogenital regions [27]. A surgical technique which can minimise the sacrifice of normal tissue and evaluate margins accurately, MMS, was developed and could provide complete microscopic margin evaluation by using the horizontal frozen histopathological sections of the entire periphery for the surgical specimens [15]. MMS is widely applied in our centre to treat the anogenital EMPD and was suggested to have longer PFS than patients who underwent WLE (HR = 0.43, 95%CI 0.21–0.87, *p* = 0.019), which is consistent with the previous study [24]. LVI was proposed as an adverse effect of CSS in anogenital EMPD patients [28]. The current study draws a similar conclusion from the univariate analysis by considering CSS using data from the centre cohort. Anogenital EMPD lesions are usually difficult to treat with surgical resection because of the potential mutilation and functional impairment caused by damaging the anogenital tissue. Various reconstruction techniques, such as free skin flaps and skin grafting [29], are employed at the wound site during surgery for anogenital EMPD to avoid excessive tension and possible complications. Surprisingly, in this study, the reconstruction of the surgical wound using free skin flaps was correlated with shorter PFS (HR = 3.58, 95%CI 1.29–9.92, *p* = 0.014) and CSS (HR = 5.49, 95%CI 1.60–18.86, *p* = 0.007) than other reconstruction modes. Gentileschi et al. considered that scar tissue from skin flaps might conceal the early signs of recurrence and impair detection [30]. More data are necessary to validate the conclusion concerning skin flaps. However, more frequent follow-up reviews and perhaps a regular biopsy in the postoperative scar for those patients who received skin flaps during reconstruction are appropriate treatment options until this correlation is investigated further.

Unlike other common tumours, there still lacks a widely accepted TNM stage system specific for EMPD because of its rarity [15]. A retrospective analysis from Japan enrolled 301 patients with EMPD. It proposed a TNM staging system which considered tumour thickness, IVL, number of metastatic lymph nodes and distant metastases as references for staging classification [31]. To some extent, this staging system verified the reliability of our study outcomes. The multivariate analysis set PFS and CCS as endpoints, and both revealed tumour invasion level and metastasis of lymph nodes as independent risk factors for poor prognosis. Invasive EMPD was defined as when the tumour penetrates the basement membrane, entering the underlying stroma instead of remaining confined to the epidermis [32]. Tumour thickness, in other words, tumour invasion level was identified as correlating significantly with CSS, with patients who experienced an invasion of EMPD into the reticular dermis or subcutaneous tissue having significantly shorter 5-year CSS than in situ in the epidermis or papillary dermis (79.1% vs. 100%, *p* < 0.0001) [21]. Although only 4.55% (8/176) of patients within our centre underwent a lymphadenectomy, similar to previous studies [31], metastases within the lymph nodes strongly correlate with a poor prognosis. Biopsy of sentinel lymph nodes (SLN) is universally adopted during breast cancer treatment. Routine biopsies are still in dispute for EMPD patients. Some studies show that EMPD patients with tumour-positive SLN have significantly prolonged survival compared to SLN-negative ones [33], and others hold an opposing view [34]. More research is necessary to verify the practicability and effectiveness of biopsy outcomes for SLN data. A routine biopsy procedure has the potential for many anogenital EMPD patients to avoid the dissection of inguinal lymph nodes, which may lead to severe complications, such as oedema of the lower extremities. Widely recognized methods and indications for the non-surgical treatment of primary EMPD are still lacking. In light of this, we need more studies concentrated on the pathogenesis of this relatively rare malignant disease so that we can find more therapeutic targets [35]. For example, hyperplasia of Toker cell has been considered to have a relationship with the occurrence of primary EMPD.

Although the current study has a large sample size compared with previous EMPD studies, there are still several limitations. Firstly, this retrospective study lacks detailed information such as radiation fields, chemotherapeutic agents, the distance between surgical margins and tumour edge and serum concentration of carcinoma embryonic antigen. Secondly, the variables provided by the SEER database and our centre are not identical, leading to the unsatisfactory matched degree for the data outcomes and results. Consequently, building a model via one of the data sources and verifying the accuracy of this model in another source may not be possible. Thirdly, selection bias exists, as in all other retrospective studies.

## 5. Conclusions

Treating anogenital EMPD, as the most common type of EMPD, has more challenges than lesions in other sites. The accurate evaluation for excision extension, the damage of structure related in the anogenital regions, high local recurrence rate and reduced quality of life have become constant issues that surgeons must address while treating anogenital EMPD. Unquestionably, a deep understanding of clinicopathological characteristics and prognostic risk factors of anogenital EMPD has potentially positive effects on optimising treatment and establishing more reasonable follow-up strategies. With more extensive research, better therapeutic strategies—including minimally invasive and functional-protection outcomes, such as MMS and SLN biopsy—may be adopted to improve clinical anogenital EMPD management and treatment.

## Figures and Tables

**Figure 1 jcm-12-00582-f001:**
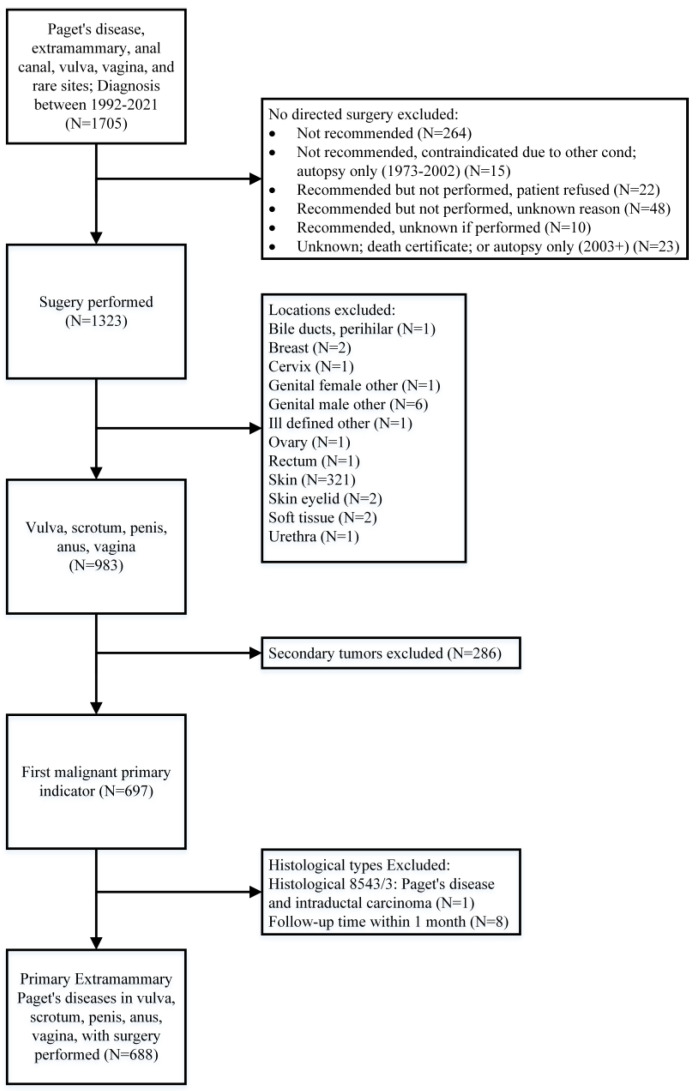
The screening process for the Surveillance, Epidemiology and End Results (SEER) program data.

**Figure 2 jcm-12-00582-f002:**
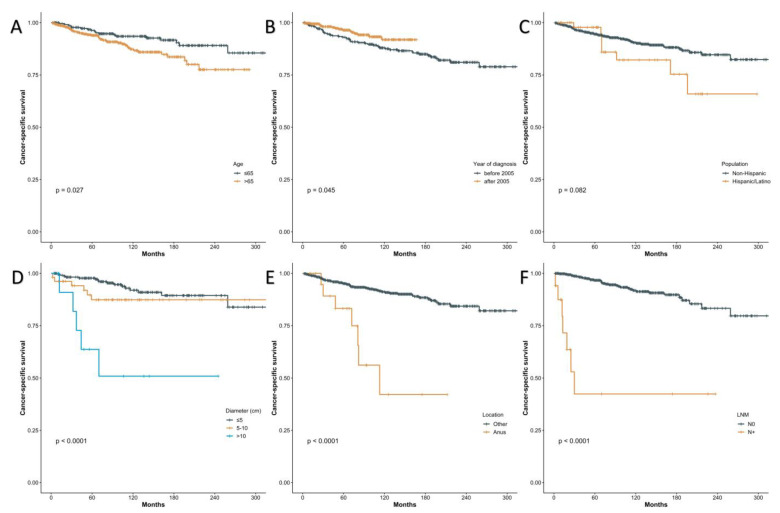
Kaplan–Meier curves of cancer-specific survival (CSS) using the SEER program data according to the age (**A**), year of diagnosis (**B**), population (**C**), diameter (**D**), location (**E**), the status of lymph nodes (**F**).

**Figure 3 jcm-12-00582-f003:**
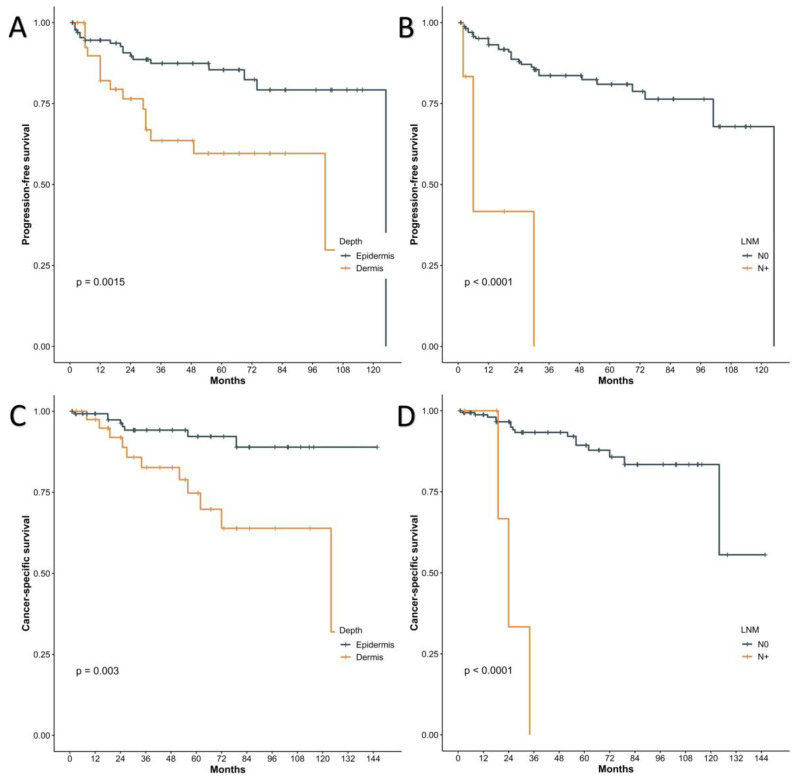
Kaplan–Meier curves of progression-free survival (PFS) using the data of our centre according to the tumour invasion depth (**A**), the status of lymph nodes (**B**). Kaplan–Meier curves of CSS using the data of our centre according to the tumour invasion depth (**C**), the status of lymph nodes (**D**).

**Table 1 jcm-12-00582-t001:** Univariate and multivariate (backward stepwise) Cox regression for cancer-specific survival (CSS) in SEER.

Variables	Univariate		Multivariate (BSR)	
	Hazard Ratio	*p* Value	Hazard Ratio	*p* Value
Age > 65 years	1.85 [1.06–3.22]	0.030	8.51 (1.67–43.42)	0.010
Male	1.29 [0.75–2.21]	0.359		
Year > 2005	0.55 [0.3–0.99]	0.048	0.30 (0.07–1.22)	0.093
Races				
White	1.00			
American Indian/Alaska Native	2.61 [0.36–18.96]	0.343		
Asian or Pacific Islander	0.67 [0.37–1.22]	0.194		
Black	0.00 [0-Inf]	0.997		
Unknown	0.00 [0-Inf]	0.998		
SEER stage				
Localized			1.00	
Regional	2.57 [1.47–4.49]	0.001	0.24 (0.04–1.61)	0.142
Distant	16.47 [6.4–42.4]	<0.001	2.93 (0.13–65.53)	0.498
N+	12.40 [5.42–28.37]	<0.001	214.24 (8.36–5493.13)	0.001
Spanish-Hispanic-Latino	1.91 [0.91–4.02]	0.087	13.44 (2.47–73.19)	0.003
Diameter (cm)				
<5				
5–10	1.76 [0.69–4.49]	0.239	2.33 (0.52–10.38)	0.268
>10	8.78 [3.18–24.23]	<0.001	14.35 (2.57–80.15)	0.002
Site				
Vulva	1.00		1.00	
Penis	0.60 [0.08–4.36]	0.614	0 (0-Inf)	0.998
Scrotum	1.00 [0.53–1.9]	0.998	2.61 (0.44–15.63)	0.293
Vagina	0.00 [0-Inf]	0.995	NA (NA-NA)	NA
Anus	5.50 [2.45–12.34]	<0.001	115.38 (6.77–1966.79)	0.001

**Table 2 jcm-12-00582-t002:** Univariate and multivariate (backward stepwise) Cox regression for PFS in EMPD.

Variables	Univariate		Multivariate (BSR)	
	Hazard Ratio	*p* Value	Hazard Ratio	*p* Value
Age > 65 years	1.15 [0.57–2.32]	0.695		
BMI > 25	1.28 [0.64–2.57]	0.488		
Female	1.99 [0.82–4.83]	0.13		
ASA > 2	0.67 [0.2–2.2]	0.505		
Comorbidity	1.22 [0.6–2.47]	0.582		
SEER stage				
Localized				
Regional	2.39 [1.16–4.94]	0.018		
Distant	4.33 [0.57–32.76]	0.156		
Diameter (cm)				
<5	1.00			
5–10	0.79 [0.37–1.68]	0.542		
>10	0.87 [0.25–3.01]	0.831		
Margin+	1.9 [0.87–4.16]	0.107		
Lymphadenectomy	2.6 [0.79–8.61]	0.117		
N+	11.66 [3.92–34.67]	<0.001	6.31 [1.91–20.88]	0.003
Invasion level				
Level 1	1.00			
Level 2	2.53 [0.85–7.54]	0.095		
Level 3	3.12 [1.46–6.67]	0.003		
Invasion level (extend through epidermis)	2.94 [1.47–5.89]	0.002	2.29 (1.06–4.98)	0.035
LVI	7.67 [2.3–25.58]	0.001	3.11 [0.82–11.86]	0.096
Chemotherapy	3.75 [0.89–15.8]	0.072		
Radiation	1.1 [0.54–2.23]	0.795		
Sites				
Vulva	1.00			
Penis	0.34 [0.07–1.71]	0.193		
Scrotum (and pubis)	0.52 [0.21–1.28]	0.157		
Reconstruction				
Direct closure	1.00	1.000		
Free skin flaps	3.58 [1.29–9.92]	0.014		
Skin grafting	2.29 [0.95–5.53]	0.066		
MMS	0.43 [0.21–0.87]	0.019		
Intervals > 2 years	2.33 [0.96–5.67]	0.062	1.00 [1.00–1.01]	0.064

**Table 3 jcm-12-00582-t003:** Univariate and multivariate (backward stepwise) Cox regression for CSS in EMPD.

Variables	Univariate		Multivariate (BSR)	
	Hazard Ratio	*p* Value	Hazard Ratio	*p* Value
Age > 65 years	1.42 [0.55–3.67]	0.469		
BMI > 25	0.97 [0.39–2.42]	0.945		
Female	0.88 [0.2–3.9]	0.87		
ASA > 2	1.24 [0.36–4.3]	0.731		
Comorbidity	0.99 [0.39–2.5]	0.979		
SEER stage				
Localized				
Regional	2.1 [0.82–5.35]	0.12		
Distant	0 [0-Inf]	0.998		
Diameter (cm)				
<5				
5–10	0.46 [0.16–1.31]	0.147		
>10	0.68 [0.15–3.06]	0.612		
Margin+	2.61 [1.02–6.65]	0.045		
N+	16.73 [4.32–64.79]	<0.001	10.33 (2.57–41.48)	<0.001
Invasion level				
Level 1	1.00			
Level 2	4.17 [1.1–15.73]	0.035		
Level 3	3.52 [1.29–9.55]	0.014		
Invasion level (extend through epidermis)	3.68 [1.47–9.25]	0.006	2.96 (1.14–7.7)	0.026
LVI	9.53 [2.11–43.09]	0.003		
Chemotherapy	3.23 [0.42–24.54]	0.258		
Radiation	0.64 [0.23–1.8]	0.397		
Sites				
Vulva	1.00			
Penis	1.34 [0.18–9.76]	0.774		
Scrotum	1.11 [0.25–4.95]	0.89		
Reconstruction				
Direct closure	1.00	1.000		
Free skin flaps	5.49 [1.6–18.86]	0.007		
Skin grafting	1.87 [0.56–6.26]	0.307		
MMS	0.48 [0.19–1.22]	0.122		
Intervals > 2 years	1.8 [0.6–5.42]	0.299		

## Data Availability

Not applicable.

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
