# Peer review of "The Clinical Characteristics and Prognostic Factors of Primary Extramammary Paget’s Disease Treated with Surgery in Anogenital Regions: A Large Population Study from the SEER Database and Our Centre"

_jcm, 2023, doi:10.3390/jcm12020582_

Round 1
Reviewer 1 Report
In this study, retrospective data of EMPD patients with anogenital region involvement and surgery are discussed.
There are no novel and original findings in the study, which presents retrospective and cross-sectional data. However, this does not mean that the data is worthless. The data presented in the study address the characteristics and demographics of EMPD patients. It analyzes with patients' clinical features and reveals statistical data. This makes the work worthwhile. I did not notice any errors in the references and tables in the study. I especially liked that the limitations were stated and the discussion section showed parallelism with the EMPD patients results.
Author Response
Comments and Suggestions for Authors: In this study, retrospective data of EMPD patients with anogenital region involvement and surgery are discussed. There are no novel and original findings in the study, which presents retrospective and cross-sectional data. However, this does not mean that the data is worthless. The data presented in the study address the characteristics and demographics of EMPD patients. It analyzes with patients' clinical features and reveals statistical data. This makes the work worthwhile. I did not notice any errors in the references and tables in the study. I especially liked that the limitations were stated and the discussion section showed parallelism with the EMPD patients results.
Response: Thanks for the reviewer’s affirmation.
Reviewer 2 Report
Today many patients with anogenital Paget are treated with Photodynamic therapy or cryotherapy. So, please specify at the title and in the discussion that these data collection are Surgery-treated patients.
Please comment and discuss at the manuscript other non-surgical modalities of treatment and possible systemic chemotherapy
Please explain features of the disease and other clinical differential diagnosis. Comment also the possible association with Toker cells.
The discussion is to long and difficult to read. Please improve
Author Response
Comments and Suggestions for Authors: Today many patients with anogenital Paget are treated with Photodynamic therapy or cryotherapy. So, please specify at the title and in the discussion that these data collection are Surgery-treated patients.
Response: We are very sorry for our negligence of the accuracy of language expression. We specify in title that the included patients were treated with surgery. The inclusion criteria in the part of data collection also clearly point out that patients with primary EMPD underwent surgery are included in our study, which are highlighted in yellow.
Comments and Suggestions for Authors: Please comment and discuss at the manuscript other non-surgical modalities of treatment and possible systemic chemotherapy.
Response: Thank you for your valuable advice. We add some content in the part of discussion. The theme of this study is about the clinical characteristics and prognostic factors of primary EMPD treated with surgery. So, we have not detailly discussed the on-surgical modalities of treatment.
Comments and Suggestions for Authors: Please explain features of the disease and other clinical differential diagnosis. Comment also the possible association with Toker cells.
Response: Thank you for your precious useful helpful suggestion. Actually, we have described features and differential diagnosis of EMPD in the part of discussion, which are highlighted in yellow. We add some content about Toker cells in the part of discussion.
Comments and Suggestions for Authors: The discussion is too long and difficult to read. Please improve.
Response: We appreciate the reviewer’s insightful suggestion. I scan the manuscript carefully. The discussion has discussed each of the valuable data according to the results. Although the discussion is too long, it is detailed and comprehensive.